# Recent Advances in Thallium Removal from Water Environment by Metal Oxide Material

**DOI:** 10.3390/ijerph20053829

**Published:** 2023-02-21

**Authors:** Xiaoyi Ren, Haopeng Feng, Mengyang Zhao, Xin Zhou, Xu Zhu, Xilian Ouyang, Jing Tang, Changwu Li, Jiajia Wang, Wangwang Tang, Lin Tang

**Affiliations:** 1College of Environmental Science and Engineering, Hunan University, Changsha 410082, China; 2Key Laboratory of Environmental Biology and Pollution Control, Ministry of Education, Hunan University, Changsha 410082, China; 3Aerospace Kaitian Environmental Technology Co., Ltd., Changsha 410100, China

**Keywords:** thallium (Tl), metal oxides, adsorption, heavy metal

## Abstract

Thallium is widely used in industrial and agricultural development. However, there is still a lack of systematic understanding of its environmental hazards and related treatment methods or technologies. Here, we critically assess the environmental behavior of thallium in aqueous systems. In addition, we first discuss the benefits and limitations of the synthetic methods of metal oxide materials that may affect the practicality and scalability of TI removal from water. We then assess the feasibility of different metal oxide materials for TI removal from water by estimating the material properties and contaminant removal mechanisms of four metal oxides (Mn, Fe, Al, and Ti). Next, we discuss the environmental factors that may inhibit the practicality and scalability of Tl removal from water. We conclude by highlighting the materials and processes that could serve as more sustainable alternatives to TI removal with further research and development.

## 1. Introduction

Thallium is a highly toxic heavy metal that can cause chronic poisoning and is widely dispersed in very low levels in the environment [1]. Thallium was originally used in medicine. It can be used to treat diseases such as ringworm of the head. It was later found to be highly toxic and used mainly in agriculture as a rodenticide, insecticide, and anti-mold agent It is currently used in large numbers, mainly in the industry [2,3]. Thallium enters the water environment mainly through the exploitation of mines containing thallium, dust deposition in ore smelting, waste discharge from sulfur mineral areas, and waste discharge from the printing and dyeing industry. It can also pollute the environment to varying degrees through the combustion of coal and sulfurous iron ore ash [4,5,6].

Studies have shown that thallium contamination has posed a serious threat to drinking water safety, and the scope of contamination has spread to countries around the world [7,8,9]. In Canada, New Brunswick, the United Kingdom, Cornwall, Idaho, China, Qianxinan, Guangdong, and Xiangnan, as well as other waters near the mine and downstream water bodies, the monitoring value of drinking water thallium exceeded 10.2–10.4 times. Even in some sections of the river, such as the Canon and the Red River in England, the single-factor pollution index for thallium has reached 14–15 times [10]. Trace amounts of thallium have also been detected in areas such as Poland [11]. The environmental ecology of these areas are also affected by thallium contamination, with levels of thallium in water bodies and plants well above background values. Thallium, along with other elements, such as arsenic, cadmium, nickel, mercury, or lead, are very harmful to mammals. However, in some specific cases, thallium is more toxic than the other elements [12]. The relocation and transport of thallium in the environment will eventually accumulate in the food chain and thus enter the human body, posing a significant threat to human health [13]. Figure 1 illustrates the impact of thallium on the environment and humans.

Thallium-containing wastewater with excessive concentrations can seriously threaten human health and environmental water quality, while traditional treatment technologies have shown significant drawbacks (e.g., poor selectivity and interference from impurity ions) [14]. Using new treatment technology to make up for the defects of traditional wastewater treatment methods has become one of the current popular research topics. In addition, there is still a lack of systematic understanding of the advantages and disadvantages of different Tl removal methods.

The current methods for thallium removal include chemical precipitation, ion exchange, solvent extraction, and adsorption [15,16,17]. The chemical precipitation method mainly removes thallium from the water by adding Cl^−^, S^2−^, potassium iron vanadium, and Prussian blue into the thallium-containing wastewater [18,19]. However, it is usually difficult for its deep treatment capability to meet the stringent thallium control standards, and there is a risk of secondary contamination. Although the ion exchange was recommended as one of the optimal treatment methods for thallium removal, the ion exchange is highly susceptible to the influence of other co-existing alkaline earth metals in thallium-containing waters. Furthermore, it is not highly selective for thallium and has the disadvantage of frequent regeneration. The solvent extraction method is usually used to recover Tl from water with high Tl concentrations, but it is difficult to apply to purifying conventional thallium-contaminated water bodies [20,21]. The adsorption method has the advantages of large adsorption capacity and high treatment depth, and it is currently widely used. Adsorption methods can be divided into physical and chemical adsorption. Physical adsorption depends on the material’s porous structure, while chemical adsorption is mainly determined by the chemical properties and state of the material surface [22]. The adsorbents currently available for thallium removal are biochar, activated carbon, metal oxides, and other materials. However, biochar [23] and activated carbon [24] showed poor performance for thallium adsorption. Recently, metal oxides have aroused wide attention due to their variable size and shape, abundant availability, sufficient surface active sites, and economic feasibility [3]. They can be used as ideal substitutes for carbon materials to remove heavy metals, and the most widely studied metal oxides include the four metal oxides of manganese, iron, aluminum, and titanium [25]. However, there are rarely studies concentrated on the feasibility and removal mechanism of metal oxide materials for TI removal from water.

Thus, this paper aims to review the progress of research on the removal of thallium from wastewater by metal oxides and to assess the feasibility of different metal oxide materials for TI removal from water by estimating the material properties and contaminant removal mechanisms of four metal oxides (Mn, Fe, Al, and Ti). Then, different environmental factors that may inhibit the practicality and scalability of Tl removal from the water will be discussed. It is hoped that this review could serve as a more sustainable alternative to TI removal with further research and development. It is also hoped that this review will provide new opportunities and possibilities for the removal and transformation of thallium to minimize the damage caused by thallium to humans and the environment.

## 2. Environmental Behavior of Thallium in Aqueous Systems

In natural water bodies, thallium is generally divided into two states: dissolved and particulate. Thallium exists mostly in its dissolved state in two forms: monovalent and trivalent thallium [26]. Tl(I) is highly soluble and mobile. In contrast, the reduction in Tl(III) is extremely fast and its solubility in water is related to the kinetics of the reaction, which is very rapid [27]. The hydrolysis of thallium is an important acid-base reaction for the generation of hydroxy complexes in the aqueous environment. It is therefore an essential factor in the de-termination of the chemical species, biological effectiveness, and toxicity of Tl in the environment [26]. In the aqueous environment, the hydrolysis of Tl(III) can be described by the following equation.
(1)$${{\mathrm{Tl}\left(\mathrm{OH}\right)}_{n-1}}^{4-n}+H_{2}O⟺{{\mathrm{Tl}(\mathrm{OH})}_{n}}^{3-n}+H^{+}$$

Solution pH and the hydrolysis constant are the main factors that determine the type of Tl in aqueous solutions. Tl(I) is soluble in water at the pH range of 3–12 to form Tl^+^, Tl(OH)_(aq)_ [28]. In the available studies, the solubility of Tl is 350 mg/L, indicating that it cannot be precipitated under normal environmental conditions. Lin and Nriagu found that at pH < 7, Tl(III) it was converted to Tl(OH)^2+^, to Tl(OH)_3_ at pH in the range 7.4–8.8, and to Tl(OH)_4_^−^ at pH 8.8 [28]. Tl can form complexes with halides (F^−^, Cl^−^, Br^−^, I^−^), sulphate ions (SO_4_^2−^), and humic acids, thus affecting the removal of Tl from the wastewater [29,30]. Tl(III)-complexes bind more strongly than Tl(I)-complexes, but Tl(I)-halides have a higher lattice energy than Tl(III)-halides. The chloride and bromide ions can inhibit the hydrolysis of Tl(III), which can form [TlCl(OH)]^+^ and [TlBr(OH)]^+^ at low halide concentrations [3]. In addition, the solubilization–precipitation behavior of Tl(III) is modified by the presence of the ligand, which becomes less oxidized in the fact of Cl^−^ (concentration of 250 mg/L).

Due to its ionic radius and electronegativity constant, Tl(I) exhibits properties similar to potassium and, to a lesser extent, to rubidium and silver in aquatic ecosystems [21]. As with alkali metal compounds, TlOH and Tl_2_O are readily soluble in water, but Tl^+^ can form strong complexes. On the other hand, monovalent thallium is similar to silver in the formation of insoluble sulfides (solubility product of 10−21.2 for Tl_2_S) and small amounts of soluble photosensitized halide complexes. Due to the large difference in the electronegativity of K^+^ and Tl^+^, monovalent thallium readily replaces potassium in clays and secondary silica. Tl(III) is mainly bound to Fe/Mn hydroxides, and thallium is adsorbed by the accumulation of mineral oxides, which are then oxidized on the mineral surface. Tl-partitioning in different sediment and suspended particle phases may provide clues to this element’s redox form in the water column.

## 3. The Synthesis and Environmental Applications of Metal Oxides

Metal oxides are binary compounds composed of oxygen and other metallic chemical elements. Metal oxides are widely used for the removal of pollutants from aqueous environmental systems because of their large specific surface area and high activity.

### 3.1. Synthesis Methods of Metal Oxides

Different methods of synthesis can lead to the different physicochemical properties of these metal oxides in different forms (e.g., manganese dioxide particles, titanium carbonate nanotubes, etc.), and differences in the specific surface area, morphology, lattice structure, size, and the zero charge point of the metal oxides can affect the mechanism of their removal of thallium [31]. The synthesis method can, therefore, impact the mechanism and performance of metal oxide removal. In past research into the removal of thallium by metal oxides, the methods of synthesis of metal oxide materials can be divided into two main types: physical and chemical methods.

#### 3.1.1. Physical Synthesis

Physical synthesis for metal oxides mainly includes high-energy ball-milling, inert gas condensation, ultrasonic blasting, severe plastic deformation, etc. The inert gas condensation method controls the particle size of the material, mainly by varying the gas pressure. Xiang et al. prepared zinc nanoparticles with different structures using an inert condensing gas method [32]. High-energy ball-milling changes the size and shape of the material by continuously grinding the material until plastic deformation occurs. Ni et al. improved the capacity and efficiency of LiMnPO_4_ by high-energy ball-milling. Ultrasonic blasting is a technique that improves the overall mechanical properties of the surface of metallic materials, resulting in a simultaneous change in the shape and dimensions of the material [33]. These methods all affect the performance of metal oxides in removing heavy metals by changing the physical characteristics of the material. For instance, the adsorption site increases as the specific surface area increases and changes in morphology result in the formation of different cavities, which facilitates the formation of surface complexes and adsorption. However, so far, there have been few studies on thallium adsorption by the physical synthesis of prepared metal oxides. Subsequent studies could focus more on thallium adsorption by the physical synthesis of metal oxides.

#### 3.1.2. Chemical Synthesis

The chemical synthesis of metal oxides for thallium removal has been relatively well studied compared to the physical synthesis. Chemical methods include controlled chemical co-precipitation, ant gelatinization (or microemulsion), chemical vapor condensation, liquid flame spraying, pulsed electrode positions, gas-phase reduction, and liquid-phase reduction [15]. Among these synthetic methods, techniques such as co-precipitation, thermal decomposition (zero-valent manganese) [34], and hydrothermal synthesis are widely used and studied due to their simplicity of operation and high yields. Li et al. used co-precipitation to synthesize a layered composite material with amorphous flower-like MnO_2_ as the outer coating and magnetite as the inner core. The mechanism of thallium removal by this material mainly occurs through the hydroxylation of the manganese dioxide surface and the formation of Mn-O surface complexes in the MnO_6_ octahedra, which are removed as Tl_2_O_3_ and Tl_2_O precipitates. In this process, Tl(I) is oxidized to Tl(III) and thus adsorbed by Mn(VI) [35]. Titanium carbonate nanotubes prepared by hydrothermal synthesis have a large specific surface area. The material removes thallium mainly by: (1) Tl^+^ exchange with Na^+^ ions; (2) Exchange of excess Tl^+^ with H^+^; and (3) Tl^+^ being oxidized to Tl^3+^ and co-precipitating with TNTs [36]. The mechanism of thallium removal by chemically synthesized metal oxides mainly involves: (i) Oxidation of Tl^+^ to Tl^3+^, thereby co-precipitating with the metal oxide; (ii) Ion exchange with ions on the metal oxide surface; (iii) Surface complexation of Tl with hydroxyl groups on the metal oxide surface; (iiii) Surface electrostatic attraction of negatively charged metal oxides to Tl^+^.

### 3.2. Removal of Thallium from Water/Wastewater by Metal Oxides

Metal oxides as adsorbents for thallium removal have the advantages of high adsorption capacity, high selectivity, and easy regeneration. The pH, co-existing ions, and organic matter affect the sorption performance and removal mechanism of thallium by different metal oxides. The studies on thallium removal by four metal oxides (manganese, aluminum, iron, and titanium) are summarized in the following section. The advantages, disadvantages, and feasibility of thallium removal by different metal oxides are analyzed to obtain the feasible method for thallium removal in real wastewater.

#### 3.2.1. Thallium Removal by Manganese Oxides

Manganese oxides are natural oxidants with good cation exchange properties and have good adsorption and oxidation properties for Tl(I). In aqueous solutions, Mn(IV) can adsorb Tl(I) by surface complexation and oxidative precipitation. At the same time, the higher valence metal in the manganese oxide can oxidize Tl(I) to form Tl_2_O_3_ precipitates, and then Tl_2_O_3_ can be absorbed on the surface of the material or in the pores [37]. The current research on thallium adsorption by manganese oxides has been carried out by four types of synthetic manganese oxides, mineral manganese oxides, modified manganese oxides, and manganese oxide composites. Reasonable modifications can improve the ability of the manganese oxides to adsorb Tl, and the selectivity and reusability of the adsorbent can be well improved. However, the lattice structure of the manganese oxides also affects the removal results. Liu et al. investigated the difference in the adsorption effect of five crystal structures of MnO_2_ (δ-, γ-, α-, β- and λ-MnO_2_) on Tl(I) [38,39]. The lattice structures of the four MnO_2_ species are shown in Figure 2. Freundlich’s isothermal adsorption curves show that δ-MnO_2_ has the maximum adsorption capacity for Tl(I) at a pH of 12 [40]. This is mainly due to the fact that the lamellar structure favors δ-MnO_2_ to carry more ions and there are different numbers of octahedral cation vacancies within its layers. These cation vacancies are considered to be strong adsorption sites for heavy metal [41]. Compared to conventional adsorbents, nanoscale adsorbents have a larger specific surface area, higher porosity, and a hollow structure, resulting in high-adsorption performance for pollutants, with the highest adsorption capacity of 672 mg/g for thallium [3]. However, nMnO_2_ is prone to coalescence in water and its separability is poor. Furthermore, there is a lack of suitable methods to remove Tl from nMnO_2_. The reproducibility of nMnO_2_ is an issue that can be further investigated. Wan et al. prepared amorphous hydrated MnO_2_ (HMO) using oxidation and chemical precipitation. Experimental results indicated that the adsorption capacity of HMO for Tl reached 352 mg/g. The mechanism of thallium adsorption by this material is mainly through the ion exchange of Mn-OH and Mn-O bonds, forming an inner sphere complex [42]. Chen et al. prepared a MnO_2_–vermiculite composite, which circumvents the problems of conventional MnO_2_ coalescence. At the same time, the specific surface area of MnO_2_–vermiculite (298.18 m^2^/g) was much larger than that of single MnO_2_, exposing more active sites and facilitating the adsorption of Tl. The Tl-O bond is formed by ion exchange on the material surface, and an oxidative precipitation process also takes place [43]. Li et al. prepared zero-valent manganese (nZVMn) in synergy with different oxidants to remove Tl from wastewater. The main mechanisms for the removal of Tl include oxidation-induced precipitation, electrostatic attraction, and surface adsorption with a removal rate of over 95.7%. It has been demonstrated that the composite material of MnO_2_ and iron can effectively remove heavy metals from water [34]. Chen et al. developed FeOOH-loaded MnO_2_ composites as emergency adsorbent materials for Tl(I) removal. Studies have shown that the Fe/Mn molar ratio of 1:2 FeOOH loaded MnO_2_ has a significantly enhanced ability to remove Tl(I) compared to pure MnO_2_, and the dissolution of MnO_2_ in this material forms new adsorption sites. Tl(I) can be oxidized to Tl(III) by this material to produce a Tl(OH)_3_ precipitate [44].

It has been found from the above studies that the predominant mechanism for thallium removal by manganese oxides is the pre-oxidation of Tl(I) to Tl(III) [40]. Subsequently, Tl(III) can either be adsorbed onto the negatively charged manganese group on the surface or generate a Tl(III) hydroxide precipitate. Tl(I) can also be adsorbed through the interaction between ions. At present, research on the removal of Tl by modified manganese oxides is still in the exploratory stage, and strengthening the selection of manganese-oxide-modified materials and modification methods can be the focus of future research. However, research cannot only be carried out under laboratory conditions but should also consider the application in real wastewater. The +1 oxidation state of thallium can be photo-oxidized (to the stable +3 state) or photo-reduced (to the elemental form). Some studies have already reported the photoelectrochemical oxidation of Tl(I) to (Tl_2_O_3_). Therefore, the effect of UV-visible light on the valence state of thallium also needs to be considered in practical water-treatment processes [45]. Finally, the manganese oxides loaded with thallium are harmful to the environment and to humans. Used manganese oxides can only be discharged into the environment once the heavy metals have been fully recovered. It is important that used sorbents are disposed of in an environmentally friendly manner and that this needs to be further investigated in subsequent research. The advantages and disadvantages of manganese oxide for thallium removal are shown in Table 1.

#### 3.2.2. Thallium Removal by Iron Oxides

Iron-oxide-based materials have advantages over other adsorbent materials due to their inherent magnetic properties, which can be easily separated after the adsorption process by applying an external magnetic field (Table 2). Previous studies have found that common iron-based materials, such as FeCl_3,_ do not adsorb thallium very well. In their study [46], Coup et al. found that the adsorption constants of hydrous iron ore for Tl^+^ could be as high as four orders of magnitude [47]. Back in 2007, Yantasee et al. found that superparamagnetic iron oxide (Fe_3_O_4_) nanoparticles functionalized with dimercaptosuccinic acid (DMSA) surfaces could effectively adsorb thallium. The results indicate that the optimized adsorption of Tl is possible at a neutral pH [48]. However, as iron oxides can undergo Fenton reactions. The disadvantages of Fenton technology are the narrow pH range (pH = 2.8–3.5), the low efficiency of iron ion recycling and the formation of undesirable iron sludge. Therefore, the modification of iron oxides is also a priority [49,50]. Solid-phase extraction methods for ferrous metal cyanides have attracted a lot of attention due to their high selectivity, rapid separation, high thermal stability, and radiation stability [51,52]. Magnetic Prussian blue (PB) was synthesized at room temperature by Zhang et al. [53]. The particles can be used for a highly selective removal of Tl. The experimental Tl removal in this study reached 528  mg/g. The high selectivity of magnetic PB for Tl is attributed to the fact that hydrated Tl has a smaller hydrated diameter and a lower hydration-free energy than other coexisting ions, and hence, can easily enter the open pores on magnetic PB. Li et al. investigated the reversible adsorption–desorption process mediated by magnetite (Fe_3_O_4_) for the removal of Tl(I) from wastewater. The results showed that adsorption under alkaline conditions achieved a rapid and effective removal of Tl(I), and desorption under acidic conditions achieved a rapid and effective enrichment of Tl(I). This study found no significant loss of magnetite in the cycling experiments, indicating that the removal and recovery of Tl(I) can be effectively and consistently repeated. This makes the reversible magnetite-based adsorption–desorption process a promising technology for further development [35].

Recently, low-cost, environmentally friendly, in situ operationally feasible, and easily magnetically selectable nano-zero-valent iron (nZVI) has started to be widely used in contaminated soil and water [54]. nZVI could activate oxygen to generate H_2_O_2_ through a two-electron reaction pathway, then H_2_O_2_ could react with Fe (II) released by nZVI, and the generated·OH could easily oxidize Tl (I) to Tl (III) (reactions 3-1 and 3-2). nZVI has a detoxifying effect on thallium-containing wastewater and groundwater, but its low adsorption capacity and rapid passivation rate limit the large-scale application of nZVI [55,56]. Therefore, Deng et al. have synthesized a nanoscale zero-valent iron, Fe@Fe_2_O_3_ core–shell nanowire (FCSN) [57]. The results showed that FCSN was able to adsorb Tl(I) with a maximum adsorption capacity of 591.7 mg/g. The material was not affected by the pH, and coexisting ions could reach a maximum adsorption rate of 95.6% when a suitable adsorbent was added. Based on the experimental data, it can be assumed that since the adsorption of Tl(I) by FCSNs is a chemical process mainly based on particle diffusion, it is not influenced by pH or coexisting ions. The material can be regenerated by desorption with H_3_PO_4_, with a desorption efficiency of 90.7%. However, the reproducibility of the material was also poor, with the removal of Tl decreasing to 52.3% after five cycles. Further studies could be explored to investigate the renewability of the material [58]. The material can be regenerated by desorption with H_3_PO_4_, with a desorption efficiency of 90.7%. However, the reproducibility of the material was also poor, with the removal of Tl decreasing to 52.3% after five cycles. Further studies could be explored to investigate the renewability of the material [59]. The mechanism of thallium adsorption by 3-ZVIMn is shown in Figure 3. The adsorption mechanism of thallium by 3-ZVIMn is shown in Figure 3. The results showed that the maximum adsorption of thallium is 990 mg/g. The Langmuir model is more appropriate for the description of thallium removal by 3-ZVIMn, indicating that the adsorption process is monolayer adsorption on a homogeneous surface. The main adsorption mechanisms are: (i) The oxidation of Tl(I) to produce Tl_2_O_3_ precipitation on the surface of the material and then be adsorbed by 3-ZVIMn; (ii) The complexation of Tl(I) with the hydroxyl group; (iii) The electrostatic attraction between the negatively charged 3-ZVIMn (pH = 10) and Tl(I). The material can be used in elution experiments with low concentrations of HCl, and the material can still adsorb 88.9% of thallium after five cycles. Liu et al. prepared amorphous zero-valent iron sulfide materials by one-step vulcanization using Na_2_S_2_O_3_ as the vulcanizing agent [56]. The S-ZVI material with S/Fe = 3 is effective in Tl(I) removal. It can directly treat high concentrations of Tl-containing wastewater with a maximum adsorption capacity of 654.4 mg/g of Tl. They also investigated the main mechanisms of thallium adsorption by S-ZVI: (i) Reaction of Tl(I) ions with sulfides in the material to form thallium sulfide precipitates (Tl_2_S); (ii) Surface complexation of Tl(I) with sulfur-iron compounds in the adsorbent; (iii) Electrostatic attraction of Tl^+^ to negatively charged S-ZVI (pH = 7). The mechanistic contribution of S-ZVI to the removal of Tl is in the order of sulfide precipitation > surface complexation > electrostatic attraction. The mechanism of thallium adsorption by S-ZVI is shown in Figure 4.
(2)$${\mathrm{Fe}(0)+O}_{2}{+2H}^{+}\;\to \mathrm{Fe}(II{)+H}_{2}O_{2}$$
(3)$$H_{2}O_{2}+\mathrm{Fe}(II)\;\to {\mathrm{OH}+\mathrm{OH}}^{-}+\mathrm{Fe}(III)$$

There is also an increasing interest in studying materials for the adsorption of thallium by combining iron oxides with other metals through modification. Fe-Mn binary oxides have the strong oxidation power of manganese dioxide and the high adsorption capacity of iron oxides. Li et al. used simultaneous chemical oxidation and the precipitation method to synthesize of Fe-Mn bimetallic materials for the removal of Tl(I). Tl adsorption to Fe-Mn adsorbents is fast, efficient, and selective. Equilibrium adsorption can be achieved by over 95% at a wide range of the operating pH values (3–12) and high ionic strengths (0.1–0.5 mol·L^−1^). This study demonstrates that the mechanism of Tl adsorption is mainly in the complexation of hydroxyl groups and Tl(I) on the Fe-Mn surface, as well as the oxidation of Tl(I) to Tl(Ⅲ) and the generation of Tl_2_O_3_ precipitates on the surface of the material to be adsorbed by the material [60]. Li et al. fabricated a graded composite containing magnetic pyrite slag and MnO_2_ to remove Tl(I) [35]. The results show that thallium is well adsorbed at a pH from 2 to 12, independent of coexisting cations and humic acids. The mechanism of Tl adsorption by this material is shown in Figure 5. In addition, after five cycles of regeneration experiments, there was no significant variation in the adsorption capacity of the Fe-Mn composites for Tl(I). Fe-Mn-binary-oxide-activated aluminosilicate minerals (FMAAM) were prepared using the hydrothermal synthesis of KMnO_4_, activated aluminosilicates, and Fe(NO_3_)_3_. The mesoporous structure of FMAAM provided a large number of surface-binding sites for Tl(I), and the maximum capacity of FMAAM for Tl(I) adsorption was 78.06 mg/g. The results of the XPS analysis indicate that the main mechanisms for the removal of Tl(I) include oxidative precipitation, ion exchange, and surface complexation [61]. Fe_3_O_4_@TiO_2_, synthesized by loading iron oxide on the surface of reduced graphene oxide (RGO) nanosheets and subsequently coated with titanium oxide, had a maximum capacity of 673.2 mg/g of Tl adsorbed at pH 8.0. The analysis results show that the -COOH group on the material present can partially oxidize Fe(II) to Fe(III). Meanwhile, the formation of more oxygen-containing functional groups on the RGO nanosheets leads to the rapid oxidation of Tl^+^ to Tl^3+^. Therefore, oxidation and precipitation are the mechanisms by which RGO nanosheets remove Tl from wastewater. The material maintains a stable Tl removal efficiency (three cycles) and does not require regeneration or activation procedures to restore its performance in the cycling test [62]. However, the cost of the material has not been considered in any of the above studies. Further research will have to consider the economic feasibility and viability of the actual wastewater treatment process.

For iron oxides, current studies have found that the mechanisms of thallium adsorption include surface complexation, electrostatic attraction, oxidative precipitation (sulfide precipitation), and ion exchange. However, few materials are unaffected by pH, co-existing ions, etc., and the reproducibility of the materials requires further discussion. Materials unaffected by pH, co-existing ions, and organic matter that have good regenerative properties are still need to be developed. Furthermore, the use of these materials in actual thallium-containing water/wastewater needs to be further verified.

**Table 2 ijerph-20-03829-t002:** Iron oxide removal thallium performance and its advantages and disadvantages.

Metal Oxides	Dosage	Thallium Concentration in the Tested Water	Performance	Advantages	Disadvantages	Ref.
Magnetic Prussian blue	0.6 g/L	1000 μg/L	528 mg/L	High removal efficiency	Poor reproducibility	[53]
Fe@Fe_2_O_3_ Core–Shell	0.75 g/L	10 mg/L	95.60%	Less influenced by environmental conditions	Poor reproducibility	[57]
3-ZVIMn	0.1 g/L	200 mg/L	990 mg/g	High removal efficiency	Highly influenced by environmental conditions	[59]
Fe-Mn binary oxides	0.5 g/L	10 mg/L	197.6 mg/g	Less influenced by environmental conditions	Poor reproducibility	[60]
Titanium iron magnetic	0.1 g/L	12.5 mg/L	111.3 mg/g	Less influenced by environmental conditions	Poor reproducibility	[58]
Fe-Mn binary oxides activated aluminosilicate mineral	1.0 g/L	10 mg/L	78.06 mg/g	Less influenced by environmental conditions	Poor removal ability	[61]
Fe_3_O_4_@TiO_2_ decorated RGO nanosheets	0.2 g/L	_	673.2 mg/g	High removal efficiency	Highly influenced by environmental conditions	[62]

#### 3.2.3. Thallium Removal by Aluminum Oxides

Activated alumina (AA), with its large surface area, pore size, and wide distribution of micropores, is a traditional adsorbent used to remove heavy metals [63]. The USEPA has recommended AA as the best available technology for Tl removal from water [64]. However, many studies have demonstrated the low efficiency of activated alumina for thallium removal [65]. The adsorption of thallium by nano-Al_2_O_3_ has attracted widespread attention. Zhang et al. chose nano-Al_2_O_3_ as the object of study for the adsorption of thallium. The results showed that the adsorption of trivalent thallium at pH = 4.5 was as high as 99.56%. This is mainly due to the fact that most atoms are not saturated and are highly susceptible to chemisorption with other atoms on the surface of the nanoparticles [66].

Aluminum composites can also be used as a method of removing thallium. Senol and Ulusoy investigated the adsorption properties of PAAm-Z and PAAm-B on Tl^+^ and Tl^3+^ by the direct polymerization of polyacrylamide (PAAm), zeolite (Z), and bentonite (B) composites (PAAm-Z, PAAm-B) prepared from AAm monomers in clay and zeolite suspensions [67]. SEM images of PAAm, PAAm-Z and PAAm-B are shown in Figure 6. The high affinity of PAAm-Z for Tl is probably due to the wide distribution of the “zeolite” particles. This leads to an increase in the specific surface area and the number of active sites for adsorption. The microporous structure formed in the material also facilitates ion exchange with Tl. The researchers performed four regeneration experiments on PAAm-Z and PAAm-B using 0.25 mol·L^−1^ HCL (the materials were used five times in total). The results of the IR spectrograms show that the efficiency of PAAm-Z decreases significantly after five uses, which may be due to blockage of the adsorption center during acid regeneration or due to the breakdown of some of the active sites in the zeolite. Other researchers have utilized aluminum beverage can powder (AlCP) as a replacement for zero-valent aluminum and the oxidation of Tl(I) at pH 9.5 followed by an alkaline-induced precipitation of Tl(III) to achieve thallium removal. The results show a maximum removal rate of 92% for thallium. The mechanism of thallium removal from water by aluminum oxides is mainly through surface complexation, ion exchange, and electrostatic attraction processes [68]. There are relatively few studies on aluminum oxides, probably due to their instability. Moreover, there are few studies on the effects of organic matter, co-existing ions, and pH on thallium removal by aluminum oxides. However, it was shown that aluminum oxide (Al_2_O_3_) with an average particle size of 63 μm was modified with the anionic surfactant sodium dodecyl sulfate (SDS) and applied to the solid-phase extraction separation of (I) and (III) thallium and the pre-enrichment of Tl(III) in wastewater. Only Tl(III) was finally absorbed left on the adsorbent, and the material was also well tolerated to Pb and Cd ions, making it a fast and effective material for Tl extraction and morphological analysis [69]. The renewable nature of the material and the practical application implications are also rarely mentioned and will be discussed further in future studies [38]. The advantages and disadvantages of aluminum oxide for thallium removal are shown in Table 3.

#### 3.2.4. Thallium Removal by Titanium Oxides

Titanium oxides can also play a role in the removal of thallium [38]. As early as 2003, Kajitvichyanukul et al. studied the titanium dioxide particles adsorbed Tl(I) in aqueous media. The results show that up to 95% of Tl(I) is bound to titanium dioxide at low-phosphate concentration levels [71]. The removal mechanism is mainly through surface complexation and electrostatic attraction. In recent years, researches have shifted to materials at the nanoscale. Zhang et al. investigated the removal of Tl(III) from water by titanium dioxide nanoparticles. They showed that at pH = 4.5, the removal of Tl^3+^ from the solution by titanium dioxide nanoparticles was close to 100%, but the experiment did not consider the regeneration of the titanium dioxide nanomaterials [72].

Titanium carbonate nanotubes (TNTs) are also some of the materials used to remove thallium from wastewater [73]. Liu et al. demonstrated that hydrothermally synthesized TNTs performed well in the adsorption of highly toxic Tl^+^/Tl^3+^ [36]. TNTs can be synthesized hydrothermally using TiO_2_ and NaOH solutions at moderate temperatures. Hydrothermally synthesized titanate nanotubes are effective in removing Tl(I) and Tl(III) from aqueous solutions because of their layered structure consisting of small diameter tubes with large surface areas. The mechanism of Tl removal by TNTs is shown in Figure 7. The structural changes of TNTs after the adsorption of Tl(I) and Tl(III) are shown in Figure 8. The maximum capacity of Tl(I) adsorbed by TNTs was 709.2 mg/g. The mechanism of the Tl(I) removal from wastewater by TNTs is mainly due to the ion exchange between Na^+^ and Tl^+^ in the interlayer of TNTs. XPS analysis showed that the removal of Tl(III) by TNTs consisted of two stages: a co-precipitation stage at high concentrations and an ion exchange stage at low concentrations. The experiment also considered the regeneration of TNTs. The results showed that TNTs also exhibit a significant adsorption capacity for thallium after the desorption by HNO_3_ and the regeneration by NaOH.

Titanium peroxide, synthesized by simple oxidation combined with precipitation, has also been used as a candidate material for the removal of thallium. Zhang et al. investigated the performance and mechanism of Tl(I) adsorption by titanium peroxide. The main adsorption process is the ionic exchange of the hydroxyl groups on the surface of titanium peroxide with Tl(I) to form Ti-O-Tl complexes. The results showed that Tl(I) adsorption increased with increasing pH. The maximum capacity of Tl(I) adsorbed by titanium peroxide at pH = 7 was 412 mg/g, indicating that titanium peroxide was very effective in removing Tl(I) from water. The mechanism is shown in Figure 9 [74]. However, the material is subjected to a maximum of three adsorption–regeneration cycles, HNO_3_ is not suitable for regenerating titanium peroxide.

The adsorption of thallium by titanium oxides is mainly accomplished by oxidative precipitation, ion exchange reactions, and electrostatic attraction. The most used in current research is titanium carbonate nanotubes. However, this is due to the high cost of titanium carbonate nanotubes, the influence of organic matter, co-existing ions, and pH on it. Therefore, the above issues need to be considered in subsequent studies.

The removal of thallium from water using metal oxides has been extensively studied, but most of them were studied based on laboratory conditions. As thallium is present in much lower concentrations in actual wastewater than under laboratory conditions, the feasibility of the studied methods for thallium removal of thallium in actual wastewater needs to be further investigated. Only methods with reasonable costs and high removal efficiency are widely used in the actual treatment of water/wastewater. For sustainability reasons, the reuse of materials should also be of concern. In the published literature, only a few studies have been carried out on material recyclability, and materials with low recyclability are not suitable for practical wastewater removal from thallium. The exhausted material is enriched with large amounts of thallium, which can easily cause secondary contamination, but there is very little mention of how to deal with the depleted material in current research. It is necessary to consider this in subsequent studies. Table 4 shows the advantages and disadvantages of titanium oxide for thallium removal.

### 3.3. Factors Affecting the Effectiveness of Thallium Removal by Metal Oxides

The main factors affecting the effectiveness of metal oxides in removing thallium from aqueous systems are pH, co-existing ions, and organic matter. Therefore, we analyzed these three factors in the removal of thallium by metal oxides, summarized some conclusions, identified some issues that can be addressed in future research, and enabled the better application of metal oxides for thallium removal.

#### 3.3.1. Effect of pH

The surface charge of metal oxides is significantly influenced by the pH of the solution. When the pH < pH_PZC_, H^+^ on the adsorbent surface would prevent the Tl(I) adsorption by electrostatic repulsion. Possible ion exchange between Tl(I) and the hydroxyl group on the adsorbent surface contributes to the removal of Tl. In addition, Tl(I) can also be oxidized to Tl(III), which then precipitates Tl(OH)_3_ on the surface of the metal oxides. As the pH of the solution increases, the positive charge on the adsorbent surface will be consumed with OH reaction. When pH > pH_PZC_, the metal oxides’ surface is negatively charged and has great affinity for positive ions, and Tl(I) could be adsorbed with high affinity through electrostatic attraction. The adsorption mechanisms for Tl(I) at low and high pH values are shown in Figure 10 [21].

The effect of pH is different for different metal oxides. For manganese oxides, on the one hand, pH changes the oxidation of the metal oxide and its surface charge; on the other hand, the solution contains a large amount of H^+^, which leads to the oxidation of Tl^+^, and then the generated Tl^3+^ forms sediment [21]. Redox reactions between heavy metals and manganese oxide at different pH conditions play an important role in the adsorption of Tl [75]. Previous studies have indicated that different manganese oxide adsorbent materials have different pH ranges for which they are required, but that the majority of manganese oxides are best removed under alkaline conditions. For example, the removal efficiency of Tl(I) by nano-MnO_2_ was lower under acidic conditions than under neutral and alkaline conditions [76,77]. For nanoparticles with zero-valent manganese, at pH > 12, it was observed that more than 95.7% of thallium was removed [34]. At a high pH, the negative-charge sites enhance the attraction between Tl(I) and the hydroxyl groups on the surface of FeOOH-loaded MnO_2_ composites [43]. Materials such as δ-MnO_2_ [41] and MnO_2_@slag [35] also demonstrated that their adsorption capacity reached its maximum at pH = 10.

For iron and titanium oxides, the reason for the increase in negative-charge density on the adsorbent surface with the increasing solution pH is the constant deprotonation of functional groups [78]. The enhanced electrostatic attraction between Tl(I) and the sorbent favors the adsorption of Tl(I). When the initial pH < 6, the hydroxyl groups on the surface of the titanium peroxide react with the protonate and H^+^. The ions adsorbed by forming the outer spherical complex are sensitive to ionic strength because the background electrolyte ions also form outer spherical ions via electrostatic forces. In contrast, the inner sphere complexes are less sensitive to changes in ionic strength and adsorb higher ionic strengths [74]. The adsorption of thallium by iron and titanium oxides under pH < 7 also behaves consistently with manganese oxides. Li et al. demonstrated that the removal of thallium by Fe-Mn dioxides was almost independent of pH. The removal of Fe-Mn dioxides reached more than 99% in the pH range of 5–12 [35]. For magnetite, the adsorption rate is close to 100% at pH > 12 [60]. The best adsorption results were obtained for titanium–iron magnetic adsorbents at a pH of 10 [60]. Both titanium dioxide [71] and titanium peroxide [79] reached their maximum adsorption under alkaline conditions.

It has also been documented that some materials are more conducive to Tl(I) removal under weakly acidic and weakly basic conditions. This was demonstrated, for example, by studies on the adsorption of thallium by HMO [80] and fancy MnO_2_-coated magnetic pyrite ash. In a study of titanium carbonate nanotubes it was found that thallium removal efficiency reached 100% at pH > 5 [36].

Unlike Mn, Fe, and Ti oxides, Tl(I) adsorption for Al oxides is more favorable under acidic conditions [70]. The pH affects alumina nanoparticles’ surface active-site distribution, with adsorption reactions occurring more readily at a pH of 3–4.5 [63]. The polyacrylamide–aluminosilicate reaction reaches equilibrium was much lower when the pH = 3 [67].

In summary, pH is a very important factor for thallium removal. Different metal oxides require different pH ranges, and some show good removal performance under acidic and alkaline conditions and poor performance under neutral conditions, such as perovskite nanoparticles [46]. Overall, the effect of pH on thallium removal by metal oxides is still relative. In subsequent studies, we have to consider the effect of pH and select the optimum pH as the condition for thallium removal (Table 5).

#### 3.3.2. Effect of Co-Existing Ions

The adsorption of metal ions under real water conditions is a complex process. The coexisting ions, dissolved organic matter, and many other components can occupy the active site of the adsorbent. All of these factors can reduce the effectiveness of the material for the adsorption of the target contaminant. Alkali or alkali metal ions in the water column have similar physicochemical properties to heavy metal ions, and there is competition for adsorbent sorption sites for Tl^+^. The metal oxide adsorbents were easily disturbed by coexisting ions. Lower concentrations of coexisting ions (0.001 mol·L^−1^) had less effect on the flowery MnO_2_-coated magnetic pyrite ash slag [81], γ-MnO_2_ [41], and MnO_2_@slag [35], but a coexisting ion concentration of 0.1 mol·L^−1^ resulted in a significant decrease in adsorption capacity. For manganese oxides, Ca^2+^ and Na^+^ are the most widely studied co-occurring cations affecting thallium removal efficiency. Na^+^ may weaken the electrostatic adsorption of δ-MnO_2_, with only 1 mmol·L^−1^ Ca^2+^ greatly reducing the adsorption rate of δ-MnO_2_ on thallium under medium and alkaline conditions [41]. Ca^2+^ may compete with nano-MnO_2_ for adsorption sites and affect the aggregation of nano-MnO_2_, which significantly reduces the removal of Tl [57,77]. Mg^2+^ can also affect thallium removal. In a study of zero-valent Mn nanoparticles it was found that adsorption decreased from 72% to 8.9% when Mg^2+^ existed [43]. For aluminum oxides, the effect of Ca^2+^ was the most significant. γ-Al_2_O_3_ adsorption of Tl(I) (6–26 mg/g) decreased with increasing Ca^2+^ concentration (0–5 mol·L^−1^). For nano-Al_2_O_3_, the interference of Cd^2+^, Cu^2+^, Mn^2+^, and Pb^2+^ should not be neglected [66]. In studies of iron oxides it was found that in addition to Ca^2+^ and Na^+^, which affect removal, Cu and Zn ions also compete with thallium for adsorption on copper ferricyanide at pH near neutral. For titanium oxides, the main consideration is the effect of the co-existence of K^+^, Na^+^, and Ca^2+^ ions: (1) K^+^ has similar chemical properties and ionic radii to the Tl^+^, so K^+^ interferes with the reaction between Tl(I) and the binding site; (2) The removal of Tl(I) was enhanced at low Na^+^ concentrations and slightly inhibited at high concentrations; (3) The effect of Ca^2+^ on titanium oxides is due to the different valence states and the affinity for Tl^3+^. The complexation between Ca^2+^ ions and trivalent ions is inhibited strongly due to the ionic radius and different valence states. Titanium oxide adsorbs ions by electrostatic-driven outer sphere complexation and/or cation exchange. The decrease in its adsorption capacity with increasing ionic strength is due to its sensitivity to changes in ionic strength. In addition to the above three ions, the remaining coexisting ions also affect the removal efficiency of thallium. For example, at a Cu^2+^ concentration of 10 mg/L, the adsorption capacity of the titanium–iron magnetic material for Tl decreased by approximately 43.8% [71].

However, some adsorbents are less disturbed by co-existing ions. For example, when magnetite was used to remove thallium, the adsorption of thallium by magnetite remained as high as 92% as the ionic strength increased. It is indicated that the effect of ionic strength is slight [35]. The coexisting ions Mg^2+^, PO_4_^3−^, and turbidity negatively affected the removal of Tl(I) and K_2_FeO_4_, while Ca^2+^, Na^+^, Cl^−^, HCO_3_^−^, and NO_3_^−^ had little effect on the removal of Tl(I) [70]. Co-existing ions such as Pb, Cd, and Zn also have less effect on thallium removal using Al_2_O_3_-SDS solid-phase extraction as the sorbent. The material is highly tolerant to these interfering ions [69]. Since the adsorption of Tl(I) by titanium peroxide is mainly achieved by the formation of intra-sphere complexes on its surface, the effect of ionic strength is not significant [75]. The ionic strength plays an important role in removing thallium by metal oxides. In subsequent studies, attention should be paid to the effect of competing cations such as Na^+^, Ca^2+^, and Mg^2+^ on the adsorption of Tl by the material. Although the effect of co-existing ions on thallium removal has been studied in previous studies for the above ions, in practice, however, copper, zinc, cadmium, and lead ions should be considered in studies as strong interferers/competitors for oxide retention, especially when thallium is used as tailings. We cannot only consider common co-existing ions alone in future studies but also the influence of common metal ions in tailings. Researchers can develop materials with little influence of co-existing ions to remove thallium from water/wastewater.

#### 3.3.3. Effect of Organic Matters

It is well known that natural organic matter (NOM) plays an important part in the transfer of metal ions [82]. NOM is a complex mixture of compounds. However, the most studied and reactive fraction is the humus consisting of fulvic acid (FA) and humic acid (HA). The complexation of thallium with HA increases with increasing HA concentration, indicating that thallium binds well to HA. If HA is present, thallium will bind to HA and lead to a reduction in adsorption efficiency.

The presence of organic matter affects the oxidation and removal of Tl(I). The organic matter is adsorbed on the adsorbent surface, thus affecting the contact of Tl with the adsorption sites. The functional groups in the organic matter can complex with Tl, forming stable compounds and reducing the removal rate. For manganese oxides, FA and HA contain complexing functional groups such as -COOH and -OH, which complex manganese and make it more soluble in the solution. NOM can reduce dissolved manganese oxides. EDTA and DTPA have a greater inhibitory effect on thallium adsorption than HA [35,41,63,76]. However, strong resistance to HA was demonstrated in studies of flowery MnO_2_-coated magnetic pyrite ash slag [83]. Tl removal decreases from 90% to 30% as HA concentration increases from 0 to 10 mg/L. EDTA and DTPA complex with Tl(III) and Tl(I), both of which have an inhibitory effect on Tl removal [35]. However, very few studies have investigated the effect of organic substances on the removal of aluminum and titanium oxides, and further efforts in this area are needed in future studies.

In summary, in actual wastewater, organic substances compete with thallium for adsorption. They can affect thallium removal, but only some of the studies have considered this aspect, and more consideration will have to be given to this issue in subsequent studies to achieve high thallium removal rates.

The pH, co-existing ions, and organic matter are important influences on thallium removal. Most current studies have considered the effect of pH and were able to identify the optimal pH for removal needs to be studied. However, since most of the current studies have been carried out under laboratory conditions and do not consider actual water background conditions, the effects of organic matter and co-existing ions have been ignored. The influence of the above factors should be considered to develop suitable techniques that effectively remove thallium from real wastewater.

## 4. Conclusions

Thallium pollution caused by industrial development is becoming increasingly serious and endangers people’s daily lives. Numerous studies have proven that the use of synthetic composites of metal oxides loaded on the surface of large organic materials to treat thallium contamination is one of the feasible technologies. Moreover, because of the different removal mechanisms of thallium by different metal oxides, it can even realize the repair and chemical extraction of thallium. Of the four metal oxides, modified iron oxides are the most promising metal oxides for future thallium removal from water/wastewater due to their high efficiency, low cost, and ease of recovery. However, the application of these metal oxide materials to actual thallium-contaminated water sources will be a major challenge for the future. In future research, the cost and regeneration of metal oxide materials will need to be considered, and the effects of pH, coexisting ion concentrations, and organic concentrations, as well as the actual background conditions of the wastewater, should also be taken into account. Furthermore, from a green and sustainable perspective, exploring the use of lower-cost nanometallic oxides for the treatment of Tl in wastewater, combined with the effects of organic matter, co-existing ions and pH, may be a more promising study for the future due to the current high price of nanometallic oxides.

## Figures and Tables

**Figure 1 ijerph-20-03829-f001:**
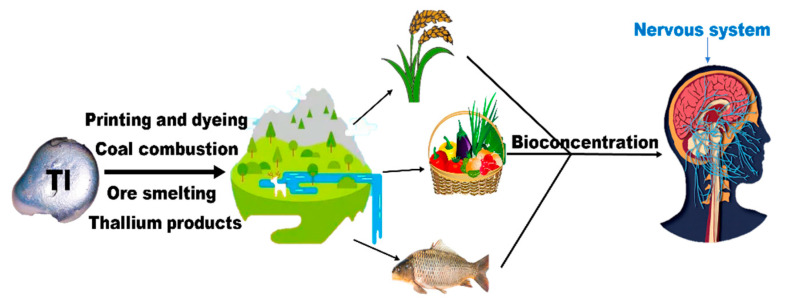
The effects of thallium entering the environment and the human nervous system.

**Figure 2 ijerph-20-03829-f002:**
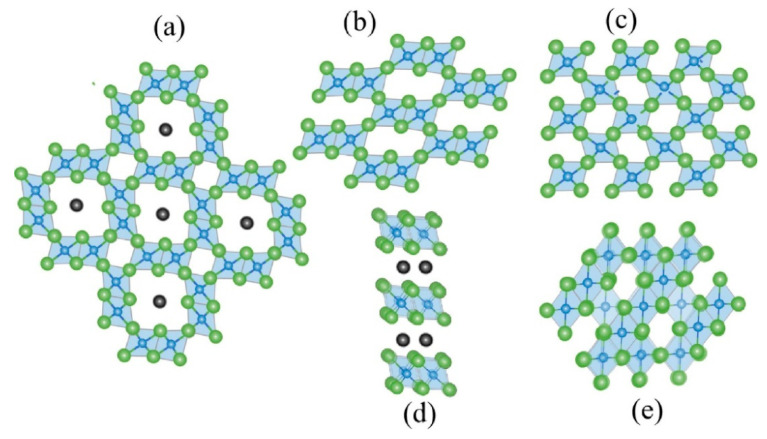
Five crystal structures of MnO_2_ (δ-, γ-, α-, β- and λ-MnO_2_): (**a**) α-MnO_2_, (**b**) γ-MnO_2_, (**c**) β-MnO_2_, (**d**) δ-MnO_2_, and (**e**) λ-MnO_2_ [40].

**Figure 3 ijerph-20-03829-f003:**
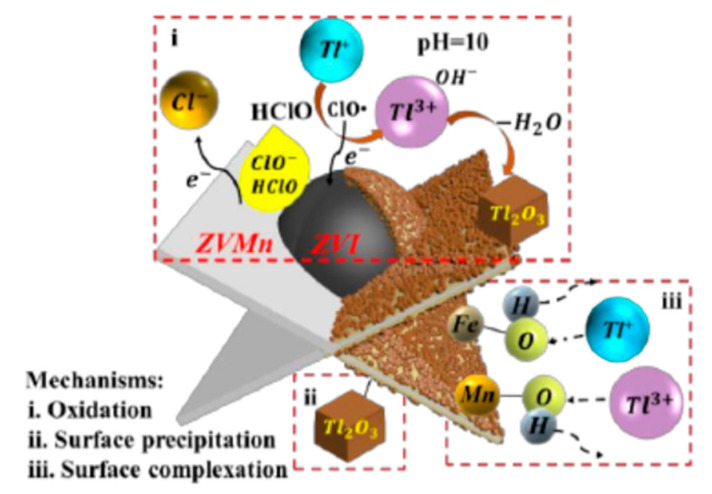
Mechanism of adsorption of thallium by 3-ZVIMn adsorbent [59].

**Figure 4 ijerph-20-03829-f004:**
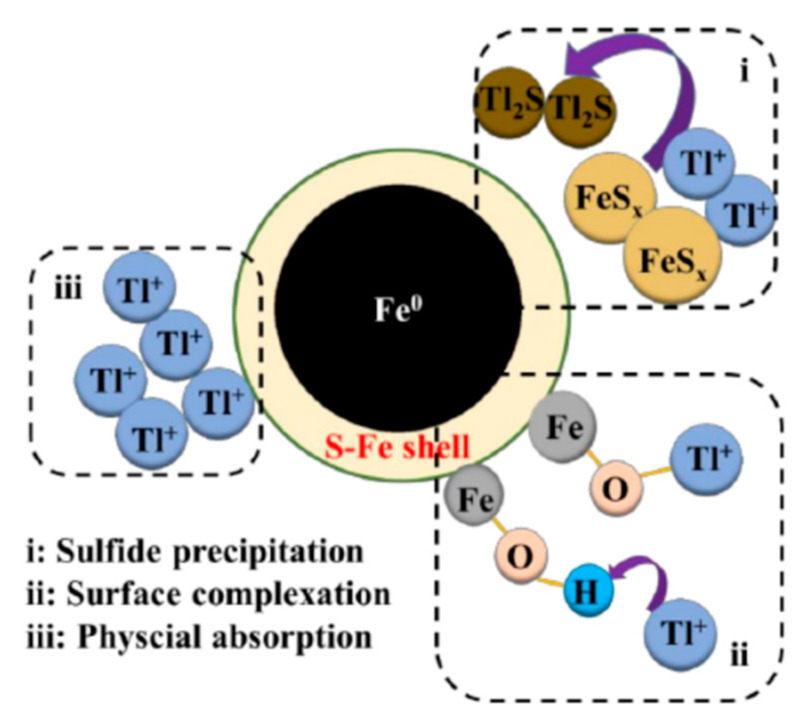
Schematic diagram of thallium adsorption mechanism of S-ZVI material [56].

**Figure 5 ijerph-20-03829-f005:**
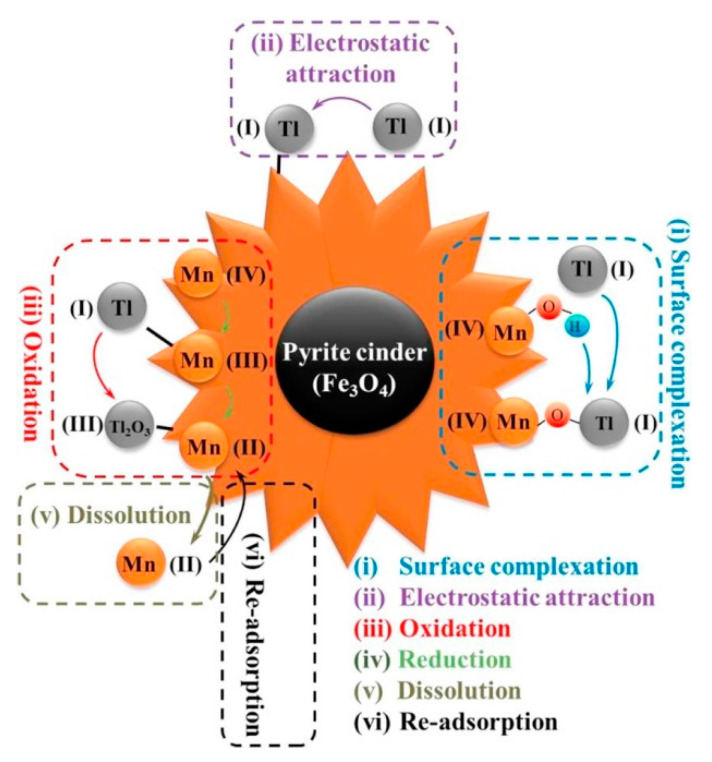
Mechanism diagram for the adsorption of Tl(I) by MnO_2_@pyrite cinder [35].

**Figure 6 ijerph-20-03829-f006:**
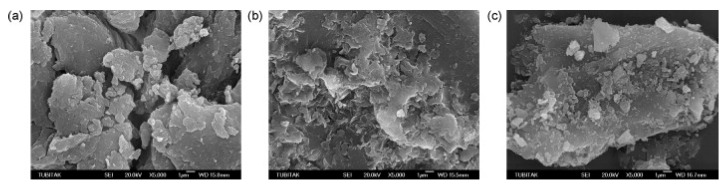
SEM views of polyacrylamide(PAAm) (**a**), polyacrylamide-bentonite(PAAm-B) (**b**), and polyacrylamide-zeolite(PAAm-Z) (**c**) [67].

**Figure 7 ijerph-20-03829-f007:**
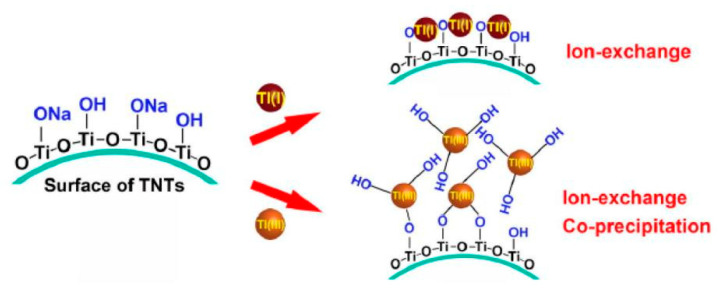
Schematic diagram of the adsorption of Tl(I) an Tl(III) by TNTs [36].

**Figure 8 ijerph-20-03829-f008:**
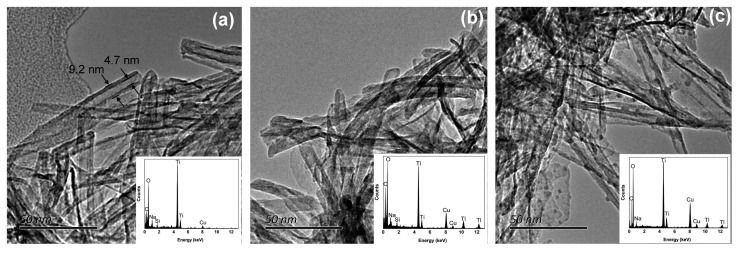
Electron micrographs of TNTs: (**a**) Before adsorption of Tl; (**b**) After absorption of thallium Tl(I); and (**c**) After absorption of thallium Tl(III) [36].

**Figure 9 ijerph-20-03829-f009:**
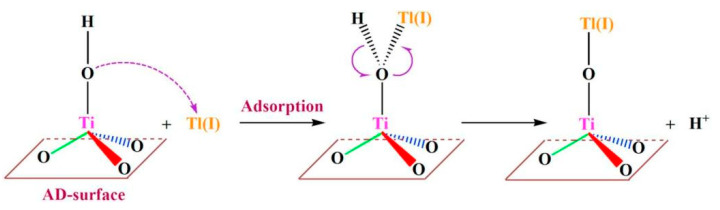
Tl removal mechanism by titanium peroxide [74].

**Figure 10 ijerph-20-03829-f010:**
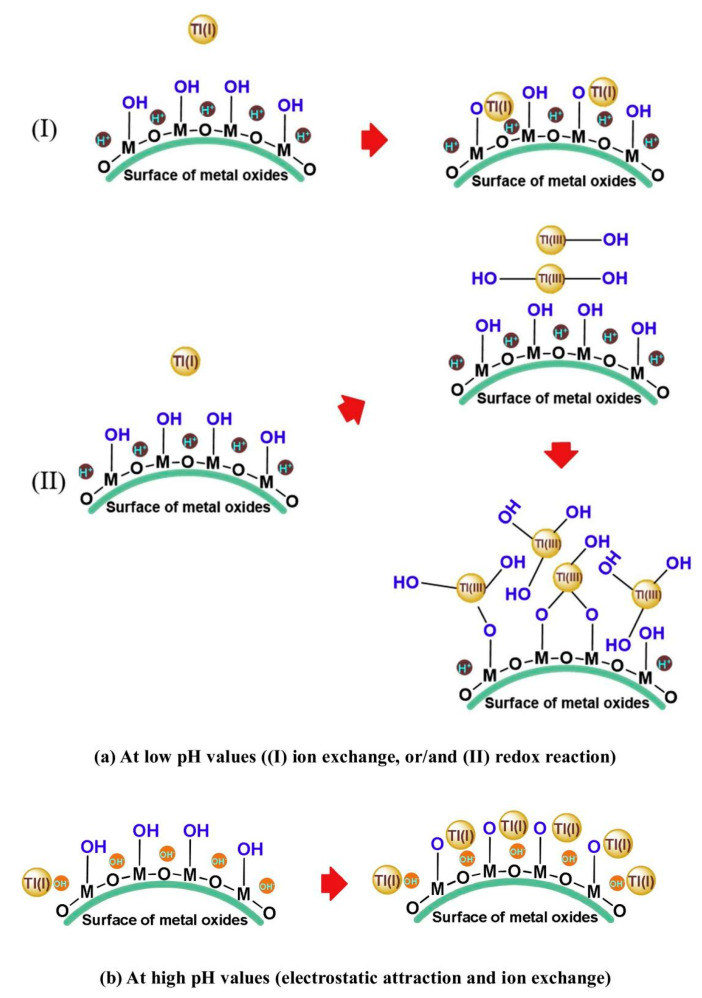
Schematic diagram of the effect of pH on the adsorption of Tl(I) by metal oxides: (**a**) At low pH; (**b**) At high pH [21].

**Table 1 ijerph-20-03829-t001:** Manganese oxide removal thallium performance and advantages and disadvantages.

Metal Oxides	Dosage	Thallium Concentration in the Tested Water	Performance	Advantages	Disadvantages	Ref.
nMnO_2_	0.05 mmol·L^−1^	3.5 mg/L	672 mg/g	High removal efficiency	Poor reproducibility	[3]
HMO	50 mg	0.5 mmol·L^−1^	348.84 mg/g	With reusable properties	Highly influenced by environmental conditions	[42]
VER-MnO_2_	—	20 μg/L	144.29 mg/g	Less influenced by environmental conditions	Poor removal ability	[43]
Zero-valent manganese nanoparticles	2 g/L	10 mg/L	95.7%	Less influenced by environmental conditions	Poor reproducibility	[34]
Flower-like manganese dioxide coated magnetic pyrite cinder	0.5 g/L	10 mg/L	320 mg/g	With reusable properties	Highly influenced by environmental conditions	[35]
FeOOH-loaded MnO_2_ nano-composite	10 mg	0~150 mg/L	450 mg/g	With reusable properties	Highly influenced by environmental conditions	[44]
δ-MnO_2_	500 μmol·L^−1^	25 mg/L	581 mg/g	High removal efficiency	Highly influenced by environmental conditions	[41]
MnO_2_@slag	1 g/L	10 mg/L	99.5%	High removal efficiency	Poor reproducibility	[35]

**Table 3 ijerph-20-03829-t003:** Aluminum oxide removal thallium performance and advantages and disadvantages.

Metal Oxides	Dosage	Thallium Concentration in the Tested Water	Performance	Advantages	Disadvantages	Ref.
Nano-Al_2_O_3_	0.03 g	10 mg/L	6.28 mg/g	Less influenced by environmental conditions	Poor reproducibility	[66]
PAAm-B	0.1 g	0.005 mol·L^−1^	197.88 mg/g-Tl(I), 32.64 mg/g-Tl(III)	High removal efficiency	Highly influenced by environmental conditions	[67]
PAAm-Z	0.1 g	0.005 mol·L^−1^	337.4 mg/g-Tl(I), 73.44 mg/g-Tl(III)	High removal efficiency	Highly influenced by environmental conditions	[67]
Ferrate pre-oxidation and poly aluminum chloride coagulation	0.5–4 mg	0.76 μg/L	87%	Less influenced by environmental conditions	Poor reproducibility	[70]

**Table 4 ijerph-20-03829-t004:** Titanium oxide removal thallium performance and its advantages and disadvantages.

Metal Oxides	Dosage	Thallium Concentration in the Tested Water	Performance	Advantages	Disadvantages	Ref.
Titanium iron magnetic	0.1 g/L	12.5 mg/L	111.3 mg/g	Less influenced by environmental conditions	Poor reproducibility	[58]
Fe_3_O_4_@TiO_2_ decorated RGO nanosheets	0.2 g/L	_	673.2 mg/g	High removal efficiency	Highly influenced by environmental conditions	[62]
TNTs	0.2 g/L	100 mg/L	709.2 mg/g	High removal efficiency	Highly influenced by environmental conditions	[36]
Titanium peroxide	0.2 g/L	0.046–20 mg/L	412 mg/g	Less influenced by environmental conditions	Poor reproducibility	[74]

**Table 5 ijerph-20-03829-t005:** Factors affecting the effectiveness of metal oxide removal.

Metal Oxides	Effect of pH Value	Effect of Co-Existing Ions	Effect of Organic Matters	Ref.
nMnO_2_	pH < 7, little influence	Ca^2+^, Mg^2+^	HA of 3 mg/L	[3]
HMO	little influence	Ca^2+^, Mg^2+^	NA	[42]
VER-MnO_2_	little influence	Ca^2+^	NA	[43]
Zero-valent manganese nanoparticles	little influence	NA	NA	[34]
Flower-like manganese dioxide coated magnetic pyrite cinder	pH < 12, little influence	Ca^2+^, Mg^2+^, Na^+^	EDTA	[35]
FeOOH-loaded MnO_2_ nano-composite	little influence	Ca^2+^	HA	[44]
δ-MnO_2_	little influence	Ca^2+^, Na^+^	NA	[41]
MnO_2_@slag	little influence	Ca^2+^	HA, FA	[35]
Nano-Al_2_O_3_	little influence	NA	HA, FA, EDTA and DTPA	[66]
PAAm-B	little influence	NA	NA	[67]
PAAm-Z	little influence	Cd(II), Cu(II), Pb(II) and Mn(II)	NA	[67]
Ferrate pre-oxidation and poly aluminum chloride coagulation	little influence	Fe^2+^, Pb^2+^, Zn^2+^	NA	[70]
Magnetic Prussian blue	little influence	NA	NA	[53]
Fe@Fe_2_O_3_ Core–Shell	NA	Mg^2+^, PO_4_^3^, Ca^2+^, Na^+^, Cl^−^	NA	[57]
3-ZVIMn	little influence	Fe^2+^, Fe^3+^	NA	[59]
Fe-Mn binary oxides	little influence	Cu^2+^, Zn^2+^	NA	[60]
Titanium iron magnetic	little influence	Na^+^, NO_3_^−^	NA	[58]
Fe-Mn-binary-oxides-activated aluminosilicate mineral	little influence	Mg^2+^, Ca^2+^	HA	[61]
Fe_3_O_4_@TiO_2_-decorated RGO nanosheets	little influence	no influence	EDTA and DTPA	[62]
TNT	little influence	Na^+^, K^+^, Ca^2+^	NA	[36]
Titanium peroxide	little influence	NA	NA	[74]

## Data Availability

The data will be made available when requested.

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
