# Peer review of "Recent Advances in Thallium Removal from Water Environment by Metal Oxide Material"

_ijerph, 2023, doi:10.3390/ijerph20053829_

Round 1
Reviewer 1 Report
Dear Author
It is important subject but in my opinion the quality of presentation is quite complicated and can be distracting to read.
The abstract is extremely general.
Page 1 – I do not agree “Thallium is more toxic … such as mercury …”
And what is “heavy metals”?
Page 2 – May be the publication such as JSS – J Soils & Sediments 5 (2) 71 – 73 (2005) (https://doi.org/10.1065/jss2005.05.136) should be added because Poland also has a region with significant contamination with this trace metal..
Page 2 chapter 2 – The trivalent thallium, its reduction is extremely fast. The solubility of Tl(III) is related to the reaction kinetics. The reaction is so fast that even in the presence of complexing ligands the reduction reached up to 10% of previous content- Electroanalysis 2014, 26, 340 – 350 (https://doi.org/10.1002/elan.201300489).
I would like to pointed that even in polluted water reservoirs the Tl concentration is below 1 ppp- JSS – J Soils & Sediments 5 (2) 71 – 73 (2005) (https://doi.org/10.1065/jss2005.05.136).
Page 3 - I do not agree that free ions so high charged –Tl3+ exists in water solution. Therefore I suggest to write Tl(III).
The previous study of contamination of Tl in any ecosystem gave some information about retention ability of Fe/MnOx and carbons minerals - Environ Chem Lett (2014) 12:435–441 (https://doi.org/10.1007/s10311-014-0468-0) or J. Hazard. Mater. 191 (2011) 170–176 (https://doi.org/10.1016/j.jhazmat.2011.04.065).
3.2.1. similar problems mentioned in chapter above.
Page 6 – the oxidation Tl(I) to Tl(III) is accelerated by UV-Vis irradiation. The problems is not discussed.
The next pages are batter in form and for reading.
3.2.3. – the surfactants e.g. SDS increased efficiency of retention of Tl and can be applied in selective retention of particular forms of Tl - Microchim Acta (2016) 183:177–183 (https://doi.org/10.1007/s00604-015-1624-3). Additionally (page 17) the surfactant effect was not consider - significant amounts of surfactants are the result of the technologies used to concentrate sulphide ores, e.g. Fe, Cu and Pb, accompanied by Tl (flotation system).
The concentration should be written mol L-1.
Page 17. – in my opinion the Cu, Zn, Cd and Pb should be consider in the study as strong interferences/ competitors for retention on oxides. The proposed the problem's solutions should be consider in the in the context of significant excesses of these metals. Tl is just tailing minerals.
Page 18 – I fully agree that the Tl mobility is not high. The properties of both speciation forms of Tl do not predispose to significant migrations in the aquatic environment. Thallium bellows to technological critical elements. But my question is “Is the described techniques has any potential to applied in remediation and chemical mining of Tl.
Author Response
Dear Editor:
Thank you very much for the letter on Feb. 7, 2023, and the valuable comments on our paper entitled " Recent advances in thallium removal from water environment by metal oxide material" (Manuscript ID: ijerph-2158207). We sincerely thank the referees for their careful review and valuable comments. The suggestions are very helpful to us and we have incorporated them into the revised paper. We checked the manuscript carefully and revised it according to the comments of the editor and reviewers. We are sure that the points listed in the checklist are addressed. Specific responses to the comments were provided one by one.
On behalf of my co-authors, I would like to provide a detailed list (point by point) of responses to each item of the comments. We also highlighted our revisions in the manuscript in red so that the editor/reviewers could easily identify them. We hope that the editor and the reviewers will be satisfied with our responses to the comments and the revisions for our original manuscript.
Thanks and best regards.
Yours sincerely,
Jiajia Wang
E-mail address: wangjiajia07@hnu.edu.cn
Reviewers’ comments
Reviewer #1:
It is important subject but in my opinion the quality of presentation is quite complicated and can be distracting to read.
Response: Thanks very much for the reviewer’s recognition and valuable comments. Based on your suggestion, we further revised this manuscript. We have made major changes in the abstract, introduction and conclusion sections. The tables in the article have been split to make the article more aesthetically pleasing to read. We have also taken on board all of the reviewers' suggestions and made changes. We hope that this revision will give you a good impression of the article.
Some issues:
- The abstract is extremely general.
Response: We really appreciate the kind suggestion. We have rewritten the abstract. The details are ‘Thallium is widely used in industrial and agricultural development. However, there is still a lack of systematic understanding of its environmental hazards and related treatment methods or technologies. Here we critically assess the Environmental behavior of thallium in aqueous systems. In addition, we first discuss the benefits and limitations of Synthetic methods of metal oxide materials that may affect the practicality and scalability of TI removal from water. We then assessed the feasibility of different metal oxide materials for TI removal from water by estimating the material properties and contaminant removal mechanisms of four metal oxides (Mn, Fe, Al, and Ti). Next, we discuss the environmental factors that may inhibit the practicality and scalability of Tl removal from water. We conclude by highlighting materials and processes that could serve as more sustainable alternatives to TI removal with further research and development.’.
- Page 1 – I do not agree “Thallium is more toxic … such as mercury …”
Response: Thanks very much for the reviewer’s valuable comments. In consideration of brevity and refinement, we used the sentence ‘Thallium is more toxic to mammals than other elements such as mercury, cadmium, lead, and zinc’ to demonstrate that thallium is highly toxic. We regret that we have not accurately expressed the magnitude of the toxicity of thallium in relation to other metals in mammals in this article, thus causing a misunderstanding among the reviewers. In response, we offer the following explanation: The U.S. EPA classifies Tl as an environmental and industrial pollutant, as well as a dangerous risk to human health and wildlife. China has included thallium in the monitoring indicator system for surface water environmental quality standards. Tl exposure can occur naturally and in occupational settings, such as the cement, electronics, and glass-manufacturing industries. Further, people can be exposed to Tl accidentally by therapeutic administration or intentionally by the ingestion of poisons. Tl is highly toxic, independent of the route of exposure, and is considered to be one of the most harmful metals to mammals. Tl is comparable to and in certain cases more toxic than, such elements as arsenic, cadmium, nickel, mercury, or lead. In summary, we will use the phrase ‘Thallium, along with other elements such as arsenic, cadmium, nickel, mercury, or lead, which is very harmful to mammals. However, in some specific cases, thallium is more toxic than the other elements.’ on page 2 instead ‘Thallium is more toxic to mammals than other elements such as mercury, cadmium, lead, and zinc.’
- And what is “heavy metals”?
Response: Many thanks to the reviewers for this question. Heavy metals are metals with a density greater than 4.5 g/cm3. In the context of environmental pollution, heavy metals mainly refer to mercury (mercury), cadmium, lead, chromium, and heavy elements of significant biological toxicity such as metal-like arsenic. This of course also includes thallium, which is studied in our article. Heavy metals are very difficult to biodegrade, but on the contrary, can be enriched thousands of times by the biomagnification of the food chain and eventually enter the human body. In the human body, heavy metals can interact strongly with proteins and enzymes, rendering them inactive, and they can also accumulate in certain organs of the body, causing chronic toxicity. Based on the reviewers' comments, we have revised the sentences of the article to ‘Thallium is a highly toxic heavy metal that can cause chronic poisoning, and widely dispersed with very low levels in the environment’.
- Page 2 – May be the publication such as JSS – J Soils & Sediments 5 (2) 71 – 73 (2005) (https://doi.org/10.1065/jss2005.05.136) should be added because Poland also has a region with significant contamination with this trace metal.
Response: We really appreciate the kind suggestion. We have included on page 2 of ‘Trace amounts of thallium have also been detected in areas such as Poland’ and cited the literature.
- Page 2 chapter 2 – The trivalent thallium, its reduction is extremely fast. The solubility of Tl(III) is related to the reaction kinetics. The reaction is so fast that even in the presence of complexing ligands the reduction reached up to 10% of previous content- Electroanalysis 2014, 26, 340 – 350 (https://doi.org/10.1002/elan.201300489).
I would like to pointed that even in polluted water reservoirs the Tl concentration is below 1 ppp- JSS – J Soils & Sediments 5 (2) 71 – 73 (2005) (https://doi.org/10.1065/jss2005.05.136).
Response: We really appreciate the kind suggestion. We have modified the ‘In contrast, Tl(III) has certain oxidation properties, has low solubility in water, and is easily precipitated. Tl(I) can be slowly reduced in the environment by Tl(III)’ to ‘In contrast, the reduction of Tl(III) is extremely fast and the solubility in water is related to the kinetics of the reaction, which is very rapid’. We also cite the article Electroanalysis 2014, 26, 340 - 350 (https://doi.org/10.1002/elan.201300489).
- Page 3 - I do not agree that free ions so high charged –Tl3+ exists in water solution. Therefore I suggest to write Tl(III).
The previous study of contamination of Tl in any ecosystem gave some information about retention ability of Fe/MnOx and carbons minerals - Environ Chem Lett (2014) 12:435–441 (https://doi.org/10.1007/s10311-014-0468-0) or J. Hazard. Mater. 191 (2011) 170–176 (https://doi.org/10.1016/j.jhazmat.2011.04.065).
Response: Many thanks for the valuable suggestion. After we carefully read the references given by the reviewers and combined them with the relevant information we reviewed, we changed the article from Tl3+ was changed to Tl(III) on page 3.
- 3.2.1. similar problems mentioned in chapter above.
Page 6 – the oxidation Tl(I) to Tl(III) is accelerated by UV-Vis irradiation. The problems is not discussed.
Response: Many thanks for the valuable suggestion. Based on the reviewers' comments, we have carefully reviewed the information and found that thallium does have photoelectrochemical oxidation capabilities, which have now been added to the article. We have added ‘The +1 oxidation state of thallium can be photo-oxidized (to the stable +3 state) or photo-reduced (to the elemental form). Some studies have already reported the photoelectrochemical oxidation of Tl(I) to (Tl2O3). Therefore, the effect of UV-visible light on the valence state of thallium also needs to be considered in practical water treatment processes’ on page 6.
- The next pages are batter in form and for reading.
Response: Many thanks to the reviewers for this question. In response to the reviewers' comments, we have reorganized the next few pages and the formatting. We have divided Table 1 into four sub-tables and placed them after each sub-section. We hope that this will give the reviewers a better impression of the paper.
- 3.2.3. – the surfactants e.g. SDS increased efficiency of retention of Tl and can be applied in selective retention of particular forms of Tl - Microchim Acta (2016) 183:177–183 (https://doi.org/10.1007/s00604-015-1624-3). Additionally (page 17) the surfactant effect was not consider - significant amounts of surfactants are the result of the technologies used to concentrate sulphide ores, e.g. Fe, Cu and Pb, accompanied by Tl (flotation system).
Response: Many thanks for the valuable suggestion. Based on the reviewer's comments, we have read this reference carefully and found that the effect of surfactants on thallium removal from aluminum oxides is relatively significant. We have made changes in the appropriate places, specifically.
(1) We have added the corresponding studies and cited references in 3.2.3. The details are ‘ However, it was shown that aluminum oxide (Al2O3) with an average particle size of 63 μm was modified with the anionic surfactant sodium dodecyl sulfate (SDS) and applied to the solid phase extraction separation of (I) and (III) thallium and the pre-enrichment of Tl(III) in wastewater. Only Tl(III) was finally adsorbed left on the adsorbent and the material was also well tolerated to Pb and Cd ions, making it a fast and effective material for Tl extraction and morphological analysis ’.
(2) Added accordingly and cited in the literature on page 18. The details are ‘ Coexisting ions such as Pb, Cd, and Zn also have less effect on thallium removal using Al2O3-SDS solid phase extraction as the sorbent. The material is highly tolerant to these interfering ions ’.
- The concentration should be written mol·L-1.
Response: Many thanks for the valuable suggestion. In response to the reviewer's comments, we have replaced all concentration units appearing in the article with mol·L-1.
- Page 17. – in my opinion the Cu, Zn, Cd and Pb should be consider in the study as strong interferences/ competitors for retention on oxides. The proposed the problem's solutions should be consider in the in the context of significant excesses of these metals. Tl is just tailing minerals.
Response: Many thanks for the valuable suggestion. Based on the reviewers' comments, we have reviewed the literature and concluded that in the actual treatment of thallium in wastewater, copper, zinc, cadmium and lead plasma do have an impact on the effectiveness of the metal oxides. We have added and cited the literature where appropriate. The details are ‘Although the effect of coexisting ions on thallium removal has been studied in previous studies for the above ions. In practice, however, copper, zinc, cadmium, and lead ions should be considered in studies as strong interferers/competitors for oxide retention, especially when thallium is used as tailings. We cannot only consider common coexisting ions alone in future studies but also the influence of common metal ions in tailings’.
- Page 18 – I fully agree that the Tl mobility is not high. The properties of both speciation forms of Tl do not predispose to significant migrations in the aquatic environment. Thallium bellows to technological critical elements. But my question is “Is the described techniques has any potential to applied in remediation and chemical mining of Tl.
Response: Many thanks for the valuable suggestion. Following the reviewer's suggestion, we have reviewed the relevant information. In general, the concentration of thallium contamination in wastewater is not very high. Low concentrations of thallium cannot be removed by adsorption. However, in our article, we conclude that metal oxides are more effective for the deep treatment of low concentrations of thallium in industrial wastewater as well as in tailings and have a wide range of applications. The remediation of actual thallium-contaminated water bodies or substrates can be carried out with inexpensive metal oxides or cheap mineral material with similar properties. Therefore we believe that these techniques described in the paper have the potential to be applied in future research for the remediation and chemical extraction of thallium. We have therefore included a reference to this in our conclusions. The details are ‘Moreover, because of the different removal mechanisms of thallium by different metal oxides, it can even realize the repair and chemical extraction of thallium’.

Reviewer 2 Report
Manuscript Title:
Recent Advances in Thallium Removal from Water Environment by Metal Oxide Material
The paper written by Xiaoyi Ren et al deliberate about the Thallium Removal from Water by Metal Oxide. The paper is written well but some representation of the data need to be revise.
1. Modify the abstract. The first sentence should motivate your study, then explain clearly and concisely what you did and at the end your main findings and its importance/impact. Specifically modify first five lines of abstract as these lines are not clear.
2. This present form of introduction part is unclear. The first two paragraphs of the introduction part need to be modify as information given is not systematic. It should be like sources of Thallium, impacts on the environment then on the human health then treatment technology. Some unnecessary information also given. Kindly remove unnecessary information from the introduction and make it concrete with specifically focusing on the present research.
3. Aims and objectives of the manuscript is not clear. Make it clear and mention the aims and objectives of the study clearly at the end of introduction.
4. Thallium impacts on the nervous system of human. It would be better if authors provide a pictorial representation of pathways of thallium into the environment and humans.
5. Table 1: Some references are old. Please try to give references from last 5 years.
6. Section 3.2.1. Thallium removal by manganese oxides.
Include in this section of table with relevant examples of Thallium removal by manganese oxides. Also, in each subsection pay attention on advantages and disadvantages of these methods, and delete general observations.
7. Section 3.2.2. Thallium removal by iron oxides: The same observations as above.
8. Section 3.2.3. Thallium removal by aluminum oxides: The same observations as above.
9. Section 3.2.4. Thallium removal by titanium oxides: The same observations as above.
10. I suggest you update your reference list by including more studies published over the last five years. For sure there are a lot of studies out there on this topic that can be used to explain the rationale of your study.
11. The conclusions should focus on the summary of the study, main findings, and possible implication.
The manuscript is not up to mark for publication in IJERPH, but it can be accepted for publication after considering the major revision. However, in my opinion for being accepted a series of aspects of the content and also of the language should substantially be improved. A significant number of grammatical and punctuation mistakes made it impossible to easily understand the content of the manuscript. In addition to the technical changes, the authors need to be very careful while editing the manuscript.
Author Response
Dear Editor:
Thank you very much for the letter on Feb. 7, 2023, and the valuable comments on our paper entitled " Recent advances in thallium removal from water environment by metal oxide material" (Manuscript ID: ijerph-2158207). We sincerely thank the referees for their careful review and valuable comments. The suggestions are very helpful to us and we have incorporated them into the revised paper. We checked the manuscript carefully and revised it according to the comments of the editor and reviewers. We are sure that the points listed in the checklist are addressed. Specific responses to the comments were provided one by one.
On behalf of my co-authors, I would like to provide a detailed list (point by point) of responses to each item of the comments. We also highlighted our revisions in the manuscript in red so that the editor/reviewers could easily identify them. We hope that the editor and the reviewers will be satisfied with our responses to the comments and the revisions for our original manuscript.
Thanks and best regards.
Yours sincerely,
Jiajia Wang
E-mail address: wangjiajia07@hnu.edu.cn
Reviewers’ comments
Reviewer #2:
The manuscript is not up to mark for publication in IJERPH, but it can be accepted for publication after considering the major revision. However, in my opinion for being accepted a series of aspects of the content and also of the language should substantially be improved. A significant number of grammatical and punctuation mistakes made it impossible to easily understand the content of the manuscript. In addition to the technical changes, the authors need to be very careful while editing the manuscript.
Response: Thanks very much for the reviewer’s recognition and valuable comments. We are very sorry that our grammatical and punctuation errors have caused you a poor reading experience. We have completely corrected the grammatical errors throughout the text. We have also checked and corrected the punctuation throughout the text. We hope that this correction will bring you a better reading experience.
- Modify the abstract. The first sentence should motivate your study, then explain clearly and concisely what you did and at the end your main findings and its importance/impact. Specifically modify first five lines of abstract as these lines are not clear.
Response: Many thanks for the valuable suggestion. We have rewritten the abstract. The details are ‘Thallium is widely used in industrial and agricultural development. However, there is still a lack of systematic understanding of its environmental hazards and related treatment methods or technologies. Here we critically assess the Environmental behavior of thallium in aqueous systems. In addition, we first discuss the benefits and limitations of Synthetic methods of metal oxide materials that may affect the practicality and scalability of TI removal from water. We then assessed the feasibility of different metal oxide materials for TI removal from water by estimating the material properties and contaminant removal mechanisms of four metal oxides (Mn, Fe, Al, and Ti). Next, we discuss the environmental factors that may inhibit the practicality and scalability of Tl removal from water. We conclude by highlighting materials and processes that could serve as more sustainable alternatives to TI removal with further research and development ’.
- This present form of introduction part is unclear. The first two paragraphs of the introduction part need to be modify as information given is not systematic. It should be like sources of Thallium, impacts on the environment then on the human health then treatment technology. Some unnecessary information also given. Kindly remove unnecessary information from the introduction and make it concrete with specifically focusing on the present research.
Response: Many thanks for the valuable suggestion. Based on the reviewers' suggestions, we have rewritten the introductory section. We have written it in the order of the sources of thallium, its impact on the environment, its impact on human health, and treatment technologies. The details are ‘Thallium is a highly toxic heavy metal that can cause chronic poisoning and is widely dispersed with very low levels in the environment. Thallium was originally used in medicine. It could be used to treat diseases such as ringworm of the head. It was later found to be highly toxic and used as a rodenticide, insecticide, and anti-mold agent, mainly in agriculture. It is currently used in large numbers, mainly in the industry. Thallium enters the water environment mainly through the exploitation of mines containing thallium, dust deposition in ore smelting, waste discharge from sulfur mineral areas, and waste discharge from the printing and dyeing industry. It can also pollute the environment to varying degrees through the combustion of coal and sulfurous iron ore ash.
Studies have shown that thallium contamination has posed a serious threat to drinking water safety, and the scope of contamination has spread to countries around the world. In Canada NewBrunswick, the United Kingdom Cornwall, Tdaho, China Qianxinan, Guangdong, Xiangnan, and other waters near the mine and downstream water bodies, the monitoring value of drinking water thallium exceeded 10.2-10.4 times. Even in some sections of the river, such as the Canon and the Red River in England, the single-factor pollution index for thallium has reached 14-15 times. Trace amounts of thallium have also been detected in areas such as Poland. The environmental ecology of these areas are also affected by thallium contamination with levels of thallium in water bodies and plants well above background values. Thallium, along with other elements such as arsenic, cadmium, nickel, mercury, or lead, which is very harmful to mammals. However, in some specific cases, thallium is more toxic than the other elements. The migration and transformation of thallium in the environment will eventually accumulate in the food chain and thus enter the human body, posing a significant threat to human health.
Thallium-containing wastewater with excessive concentrations can seriously threaten human health and environmental water quality, while traditional treatment technologies have shown significant drawbacks (e.g. poor selectivity, and interference from impurity ions). Using new treatment technology to make up for the defects of traditional wastewater treatment methods has become one of the current research hotspots. In addition, there is still a lack of systematic understanding of the advantages and disadvantages of different Tl removal methods.
The current methods for thallium removal include chemical precipitation, ion exchange, solvent extraction, and adsorption. The chemical precipitation method is mainly used to remove thallium from the water by adding Cl-, S2-, potassium iron vanadium, and Prussian blue into the thallium-containing wastewater. However, its deep treatment capability is usually difficult to meet the stringent thallium control standards, and there is a risk of secondary contamination. Although the ion exchange was recommended as one of the optimal treatment methods for thallium removal. However, the ion exchange is highly susceptible to the influence of other co-existing alkaline earth metals in thalli-um-containing waters. Furthermore, it is not highly selective for thallium and has the disadvantage of frequent regeneration. The solvent extraction method is usually used to recover Tl from water with high Tl concentrations, but it is difficult to apply to purifying conventional thallium-contaminated water bodies. The adsorption method has the advantages of large adsorption ability and high capacity for a deep treatment, and it is widely used at present. Adsorption methods can be divided into physical and chemical adsorption. Physical adsorption depends on the material’s porous structure, while chemical adsorption is mainly determined by the chemical properties and state of the material surface. The adsorbents currently available for thallium removal are biochar, activated carbon, metal oxides, and other materials. However, biochar and activated carbon showed poor performance for thallium adsorption. Recently, metal oxides have aroused wide attention due to their variable size and shape, abundant availability, sufficient surface active sites, and economic feasibility. They can be used as an ideal substitute for carbon materials to remove heavy metals and the most widely studied metal oxides include the four metal oxides of manganese, iron, aluminum, and titanium. However, there are rarely studies concentrated on the feasibility and removal mechanism of metal oxide materials for TI removal from water.
Thus, this paper aims to review the progress of research on the removal of thallium from wastewater by metal oxides and to assess the feasibility of different metal oxide materials for TI removal from water by estimating the material properties and contaminant removal mechanisms of four metal oxides (Mn, Fe, Al, and Ti). Then, different environmental factors that may inhibit the practicality and scalability of Tl removal from the water will be discussed. It is hoped that this review could serve as a more sustainable alternative to TI removal with further research and development. It is also hoped that this review will provide new opportunities and possibilities for the removal and transformation of thallium to minimize the damage caused by thallium to humans and the environment.
- Aims and objectives of the manuscript is not clear. Make it clear and mention the aims and objectives of the study clearly at the end of introduction.
Response: Many thanks for the valuable suggestion. Based on the reviewers' comments, we found that at the end of the introduction we did not clearly state the research aims and objectives of the review. We have re-added our research aims and objectives. The details are ‘Thus, this paper aims to review the progress of research on the removal of thallium from wastewater by metal oxides and to assess the feasibility of different metal oxide materials for TI removal from water by estimating the material properties and contaminant removal mechanisms of four metal oxides (Mn, Fe, Al, and Ti). Then, different environmental factors that may inhibit the practicality and scalability of Tl removal from the water will be discussed. It is hoped that this review could serve as a more sustainable alternative to TI removal with further research and development. It is also hoped that this review will provide new opportunities and possibilities for the removal and transformation of thallium to minimize the damage caused by thallium to humans and the environment.
- Thallium impacts on the nervous system of human. It would be better if authors provide a pictorial representation of pathways of thallium into the environment and humans.
Response: Many thanks for the valuable suggestion. In response to the reviewers' comments, we have added illustrations of thallium entering the environment and affecting the human nervous system.Please see the attached content for pictures.
- Table 1: Some references are old. Please try to give references from last 5 years.
Response: Many thanks for the valuable suggestion. In response to the reviewers' comments, we have split Table 1. It was re-listed in each subsection and the references were replaced in each section. Please refer reviewers to the responses to the next comments for details.
- Section 3.2.1. Thallium removal by manganese oxides.
Include in this section of table with relevant examples of Thallium removal by manganese oxides. Also, in each subsection pay attention on advantages and disadvantages of these methods, and delete general observations.
Response: Many thanks for the valuable suggestion. In response to the reviewers' comments, we have restated Table 1 and updated the references. In Table 1 we have indicated the advantages and disadvantages of each manganese oxide and removed any information that is not required. Please see the table below for details.
Table 1 Manganese oxide removal thallium performance and advantages and disadvantages
Metal oxides |
Dosage |
Thallium concentration in the tested water |
Performance |
Advantages |
Disadvantages |
Ref. |
|
nMnO2 |
0.05 mmol·L-1 |
3.5 mg/L |
672 mg/g |
High removal efficiency |
Poor reproducibility |
Xu et al..(Xu et al., 2019) |
|
HMO |
50 mg |
0.5 mmol·L-1 |
348.84 mg/g |
With reusable properties |
Highly influenced by environmental conditions |
Wan et al.(Wan et al., 2014) |
|
VER-MnO2 |
— |
20μg/L |
144.29 mg/g |
Less influenced by environmental conditions |
Poor removal ability |
Chen et al. (Chen et al., 2019) |
|
Zero-valent manganese nanoparticles |
2 g/L |
10 mg/L |
95.7% |
Less influenced by environmental conditions |
Poor reproducibility |
Li et al.(Li, K.K. et al., 2020) |
|
Flower-like manganese dioxide-coated magnetic pyrite cinder |
0.5 g/L |
10 mg/L |
320 mg/g |
With reusable properties |
Highly influenced by environmental conditions |
Li et al.(Li et al., 2018) |
|
FeOOH-loaded MnO2 nano-composite |
10 mg |
0~150 mg/L |
450 mg/g |
With reusable properties |
Highly influenced by environmental conditions |
Chen et al.(Chen et al., 2017) |
|
δ-MnO2 |
500 μmol·L-1 |
25 mg/L |
581 mg/g |
High removal efficiency |
Highly influenced by environmental conditions |
Song et al.(Song et al., 2020) |
|
MnO2@slag |
1 g/L |
10 mg/L |
99.50% |
High removal efficiency |
Poor reproducibility |
Li et al. (Li et al., 2018) |
- Section 3.2.2. Thallium removal by iron oxides: The same observations as above.
Response: Many thanks for the valuable suggestion. In response to the reviewers' comments, we have restated Table 2 and updated the references. In Table 2 we have indicated the advantages and disadvantages of each iron oxide and removed any information that is not required. Please see the table below for details.
Table 2 Iron oxide removal thallium performance and advantages and disadvantages.
Metal oxides |
Dosage |
Thallium concentration in the tested water |
Performance |
Advantages |
Disadvantages |
Ref. |
Magnetic Prussian blue |
0.6 g/L |
1000 μg/L |
528 mg/L |
High removal efficiency |
Poor reproducibility |
Zhang et al. (Zhang et al., 2022) |
Fe@Fe2O3 Core–Shell |
0.75 g/L |
10 mg/L |
95.60% |
Less influenced by environmental conditions |
Poor reproducibility |
Deng et al. (Deng et al., 2016) |
3-ZVIMn |
0.1 g/L |
200 mg/L |
990 mg/g |
High removal efficiency |
Highly influenced by environmental conditions |
Li et al. (Li, Y.T. et al., 2020) |
Fe-Mn binary oxides |
0.5 g/L |
10 mg/L |
197.6 mg/g |
Less influenced by environmental conditions |
Poor reproducibility |
Li et al. (Li et al., 2017) |
Titanium iron magnetic |
0.1 g/L |
12.5 mg/L |
111.3 mg/g |
Less influenced by environmental conditions |
Poor reproducibility |
Tang et al. (Tang et al., 2017) |
Fe-Mn binary oxides activated aluminosilicate minera |
1.0 g/L |
10 mg/L |
78.06 mg/g |
Less influenced by environmental conditions |
Poor removal ability |
Zou et al. (Zou et al., 2021) |
Fe3O4@TiO2 decorated RGO nanosheets |
0.2 g/L |
_ |
673.2 mg/g |
High removal efficiency |
Highly influenced by environmental conditions |
Li et al. (Li et al., 2021) |
- Section 3.2.3. Thallium removal by aluminum oxides: The same observations as above.
Response: Many thanks for the valuable suggestion. In response to the reviewers' comments, we have restated Table 3. However, we are sorry that we have not found an updated article due to the low number of articles that have studied thallium removal by aluminum oxides in recent years. However, we have added the advantages and disadvantages of each aluminum oxide in Table 3.
Table 3 Aluminum oxide removal thallium performance and advantages and disadvantages.
Metal oxides |
Dosage |
Thallium concentration in the tested water |
Performance |
Advantages |
Disadvantages |
Ref. |
|
|
Nano-Al2O3 |
0.03 g |
10 mg/L |
6.28mg/g |
Less influenced by environmental conditions |
Poor reproducibility |
Zhang et al.(Zhang et al., 2008) |
|
|
PAAm-B |
0.1 g |
0.005 mol·L-1 |
197.88 mg/g- Tl(I),32.64 mg/g - Tl(â…¢) |
High removal efficiency |
Highly influenced by environmental conditions |
Senol, Ulvi Ulusoy(Senol and Ulusoy, 2010) |
||
PAAm-Z |
0.1 g |
0.005 mol·L-1 |
337.4 mg/g -Tl(I),73.44 mg/g - Tl(â…¢) |
High removal efficiency |
Highly influenced by environmental conditions |
Senol, Ulvi Ulusoy(Senol and Ulusoy, 2010) |
||
Ferrate pre- |
0.5-4 mg |
0.76 μg/L |
87% |
Less influenced by environmental conditions |
Poor reproducibility |
Liu et al. (Liu et al., 2019) |
- Section 3.2.4. Thallium removal by titanium oxides: The same observations as above.
Response: Many thanks for the valuable suggestion. In response to the reviewers' comments, we have restated Table 4 and updated the references. In Table 4 we have indicated the advantages and disadvantages of each titanium oxide and removed any information that is not required. Please see the table below for details.
Table 4 Titanium oxide removal thallium performance and advantages and disadvantages.
Metal oxides |
Dosage |
Thallium concentration in the tested water |
Performance |
Advantages |
Disadvantages |
Ref. |
Titanium iron magnetic |
0.1g/L |
12.5 mg/L |
111.3 mg/g |
Less influenced by environmental conditions |
Poor reproducibility |
Tang et al.(Tang et al., 2017) |
Fe3O4@TiO2 decorated RGO nanosheets |
0.2g/L |
_ |
673.2 mg/g |
High removal efficiency |
Highly influenced by environmental conditions |
Li et al.(Li et al., 2021) |
TNTs |
0.2g/L |
100 mg/L |
709.2 mg/g |
High removal efficiency |
Highly influenced by environmental conditions |
Liu et al.(Liu et al., 2014) |
Titanium peroxide |
0.2g/L |
0.046-20 mg/L |
412 mg/g |
Less influenced by environmental conditions |
Poor reproducibility |
Zhang et al.(Zhang et al., 2018) |
- I suggest you update your reference list by including more studies published over the last five years. For sure there are a lot of studies out there on this topic that can be used to explain the rationale of your study.
Response: Many thanks for the valuable suggestion. As suggested by the reviewers, we have updated the reference list to include more studies on thallium removal by metal oxides over the last five years. However, after our review of the literature, we found relatively few studies on thallium removal by metal oxides in the last five years and we only found a fraction of articles that fit our research theme. We have added all of these articles to the review. We will list the references we have added below.
- Zhang HL, Qi JY, Liu F et al. (2022) One-pot synthesis of magnetic Prussian blue for the highly selective removal of thallium(I) from wastewater: Mechanism and implications Journal of Hazardous Materials 423 doi:10.1016/j.jhazmat.2021.126972
- Liu J, Luo XW, Sun YQ et al. (2019) Thallium pollution in China and removal technologies for waters: A review Environment International 126:771-790 doi:10.1016/j.envint.2019.01.076
- Zhao Z, Xiong YH, Cheng XK et al. (2020) Adsorptive removal of trace thallium(I) from wastewater: A review and new perspectives Journal of Hazardous Materials 393 doi:10.1016/j.jhazmat.2020.122378
- Liu J, Ren SX, Cao JL et al. (2021) Highly efficient removal of thallium in wastewater by MnFe2O4-biochar composite Journal of Hazardous Materials 401 doi:10.1016/j.jhazmat.2020.123311
- Chen KY, Tzou YM, Hsu LC et al. (2022) Oxidative removal of thallium(I) using Al beverage can waste with amendments of Fe: Tl speciation and removal mechanisms Chemical Engineering Journal 427 doi:10.1016/j.cej.2021.130846
- Zhang LJ, Yang Y, Wu SX et al. (2022b) Insights into the synergistic removal mechanisms of thallium(I) by biogenic manganese oxides in a wide pH range Science of the Total Environment 831 doi:10.1016/j.scitotenv.2022.154865
- Chen WP, Huangfu XL, Xiong JM et al. (2022b) Retention of thallium(I) on goethite, hematite, and manganite: Quantitative insights and mechanistic study Water Research 221 doi:10.1016/j.watres.2022.118836
- Chen WP, Xiong JM, Liu JC et al. (2022c) Thermodynamic and kinetic coupling modeling for thallium(I) sorption at a heterogeneous titanium dioxide interface Journal of Hazardous Materials 428 doi:10.1016/j.jhazmat.2022.128230
- Wang NN, Su ZB, Deng NR et al. (2020) Removal of thallium(I) from aqueous solutions using titanate nanomaterials: The performance and the influence of morphology Science of the Total Environment 717 doi:10.1016/j.scitotenv.2020.137090
- Liu YL, Zhang J, Huang HM et al. (2019b) Treatment of trace thallium in contaminated source waters by ferrate pre-oxidation and poly aluminium chloride coagulation Separation and Purification Technology 227 doi:10.1016/j.seppur.2019.06.001
- Yao JN, Wang HN, Ma CX et al. (2022) Cotransport of thallium(I) with polystyrene plastic particles in water-saturated porous media Journal of Hazardous Materials 422 doi:10.1016/j.jhazmat.2021.126910
- Soltani R, Marjani A, Shirazian S (2019) Facile one-pot synthesis of thiol-functionalized mesoporous silica submicrospheres for Tl(I) adsorption: Isotherm, kinetic and thermodynamic studies Journal of Hazardous Materials 371:146-155 doi:10.1016/j.jhazmat.2019.02.076
- Khan FSA, Mubarak NM, Tan YH et al. (2021) A comprehensive review on magnetic carbon nanotubes and carbon nanotube-based buckypaper for removal of heavy metals and dyes Journal of Hazardous Materials 413 doi:10.1016/j.jhazmat.2021.125375
- Liu F, Li H, Liao D et al. (2020) Carbon quantum dots derived from the extracellular polymeric substance of anaerobic ammonium oxidation granular sludge for detection of trace Mn(vii) and Cr(vi) RSC Advances 10:32249-32258 doi:10.1039/D0RA06133F
- Ma CX, Huang RX, Huangfu XL et al. (2022) Light- and H2O2-Mediated Redox Transformation of Thallium in Acidic Solutions Containing Iron: Kinetics and Mechanistic Insights Environmental Science & Technology 56:5530-5541 doi:10.1021/acs.est.2c00034
- Rinklebe J, Shaheen SM, El-Naggar A et al. (2020) Redox-induced mobilization of Ag, Sb, Sn, and Tl in the dissolved, colloidal and solid phase of a biochar-treated and un-treated mining soil Environment International 140 doi:10.1016/j.envint.2020.105754
- The conclusions should focus on the summary of the study, main findings, and possible implication.
Response: Many thanks for the valuable suggestion. In response to the reviewers' comments, we have rewritten the conclusions. The details are ‘Thallium pollution caused by industrial development is becoming increasingly serious and endangers people's daily lives. Numerous studies have proven that the use of synthetic composites of metal oxides loaded on the surface of large organic materials to treat thallium contamination is one of the feasible technologies. Moreover, because of the different removal mechanisms of thallium by different metal oxides, it can even realize the repair and chemical extraction of thallium. Of the four metal oxides, modified iron oxides are the most promising metal oxides for future thallium removal from wa-ter/wastewater due to their high efficiency, low cost and ease of recovery. However, the application of these metal oxide materials to actual thallium contaminated water sources will be a major challenge for the future. In future research, the cost and regeneration of metal oxide materials will need to be considered, and the effects of pH, co-existing ion concentrations and organic concentrations, as well as the actual background conditions of the wastewater, should also be taken into account. Furthermore, from a green and sustainable perspective, exploring the use of lower cost nanometallic oxides for the treatment of Tl in wastewater, combined with the effects of organic matter, co-existing ions and pH, may be a more promising study for the future due to the cur-rent high price of nanometallic oxides’.

Round 2
Reviewer 2 Report
The authors have revised the manuscript as per the suggestions given. The manuscript can be accepted in its current form.